# Overexpression of Fibroblast Growth Factor 8 Is a Predictor of Impaired Survival in Esophageal Squamous Cell Carcinoma and Correlates with ALK/EML4 Alteration

**DOI:** 10.3390/cancers16213624

**Published:** 2024-10-27

**Authors:** Gerd Jomrich, Dagmar Kollmann, Winny Yan, Daniel Winkler, Matthias Paireder, Lisa Gensthaler, Hannah Christina Puhr, Aysegül Ilhan-Mutlu, Reza Asari, Sebastian F. Schoppmann

**Affiliations:** 1Department of General Surgery, Medical University of Vienna and Gastroesophageal Tumor Unit, Comprehensive Cancer Center (CCC), 1090 Vienna, Austria; gerd.jomrich@meduniwien.ac.at (G.J.); dagmar.kollmann@meduniwien.ac.at (D.K.); winny.yan@meduniwien.ac.at (W.Y.); matthias.paireder@meduniwien.ac.at (M.P.); lisa.gensthaler@meduniwien.ac.at (L.G.); 2Institute for Retailing and Data Science, Vienna University of Economics and Business, 1020 Vienna, Austria; daniel.winkler@wu.ac.at (D.W.); reza.asari@meduniwien.ac.at (R.A.); 3Department of Medicine 1, Medical University of Vienna and Gastroesophageal Tumor Unit, Comprehensive Cancer Center (CCC), 1090 Vienna, Austria; hannah.puhr@meduniwien.ac.at (H.C.P.); aysegul.ilhan@meduniwien.ac.at (A.I.-M.)

**Keywords:** esophageal squamous cell carcinoma, upfront resection, fibroblast growth factor 8, anaplastic lymphoma kinase (ALK), microtubule-associated protein-like 4 gene (EML4), prognostic factor

## Abstract

This study investigates the role of three specific proteins—FGF8, ALK, and EML4—in predicting the prognosis of patients with esophageal squamous cell carcinoma (ESCC), a common type of esophageal cancer. The aim of the study was to assess whether the presence and levels of these proteins could indicate which patients are at a higher risk of poor survival following surgery. By analyzing tissue samples from 122 patients, who underwent surgical removal of their cancer, the study discovered that an increase in FGF8 protein levels is associated with a reduced chance of survival. Additionally, the study found a significant correlation between FGF8 and alterations in the ALK and EML4 proteins. These results suggest that FGF8 could serve as a valuable marker for predicting patient outcomes and might also become a target for future therapies aimed at improving survival rates in ESCC patients.

## 1. Introduction

With over 572,000 new cases and 509,000 deaths per year, esophageal cancer is the sixth most frequent cancer in the world and the seventh most common cause of cancer-related death [1]. Esophageal squamous cell carcinoma (ESCC) is the most common subtype of esophageal cancer, accounting for nearly 90% of cases [2]. Due to the lack of reliable biomarkers for screening, many ESCC patients do not receive an early diagnosis, meaning distant metastases have frequently already developed, significantly worsening the prognosis. There is currently no viable course of action, and there are no reliable diagnostic and prognostic biomarkers. Less than 30% of ESCC patients survive for five years [2,3]. Therefore, it is vital to find more accurate biomarkers for ESCC, as doing so will boost the effectiveness of both diagnosis and treatment as well as our comprehension of the pathogenic mechanisms.

Fibroblast growth factors (FGFs) are peptide-like molecules that control cell growth by attaching to receptors on cell membranes (FGFRs). Numerous types of tissue include FGFs, which are known to encourage the proliferation of fibroblasts. The term “heparin-conjugate growth factors” refers to FGFs with a strong affinity for heparin. The heparin sulfate (HS) domain and the fibroblast growth factor receptor-binding domain are both present in the molecular structure of FGF proteins. A total of 22 distinct FGFs are now recognized in mammals. One endocrine and one intracellular FGF subfamily exist in addition to the largest FGF subfamily, the canonical FGFs, and all FGFs mediate their cellular response by binding to and activating one of four FGFRs [4,5].

In various cancers, including adenocarcinomas of the gastroesophageal junction, the prognostic value of FGF8, a member of the FGF subfamily 8, has already been examined [6,7,8]. Additionally, there is growing evidence that fibroblast growth factor receptor (FGFR) inhibitors and FGF ligand traps can be used to overcome chemoresistance in a variety of malignancies [9,10,11].

Anaplastic lymphoma kinase (ALK), an enzyme encoded by the ALK gene, plays a pivotal role in cellular communication in humans and is also known as ALK tyrosine kinase receptor or CD246. Genetic code or regulatory DNA sequences may be exchanged between two genes as a result of gene fusion. Research on cancer has demonstrated the significance of the translated products of gene fusion. A protein can result from gene fusion, as a result of the fusion of portions of two distinct genes. EML4 (echinoderm microtubule-associated protein-like 4)-ALK in lung cancer is an example of a well-known gene fusion involving a kinase-coding gene [12,13]. In ESCC, where fusion protein tropomyosin 4 (TPM4)-ALK was found in two distinct proteomics-based studies, similar cases were also noted [14,15]. It is necessary to investigate these ALK-related gene fusion events to determine their precise role and importance in relation to ESCC in various populations. However, ALK inhibitors were not formally evaluated in the context of ESCC beyond the findings from basic science. While the data on the use of ALK inhibitors in ESCC are limited, five ALK inhibitors have been approved for the treatment of non-small-cell lung cancer with an ALK rearrangement, and while the outlook has significantly improved, the emergence of drug resistance remains a significant challenge.

Multi-kinase TKIs, such as nintedanib, have already been studied in non-small-cell lung cancer, showing activity against FGF, and target several important angiogenic pathways [16]. Furthermore, it was discovered only recently that FGFR3 activation and overexpression resulted in ALK-TKI resistance via a bypass pathway [17].

Little is currently known about the function of FGF8 in ESCC, and to the best of our knowledge, no data have ever been published describing the potential correlation with ALK/EML4 in ESCC. In order to define predictive markers and possibly find suitable new targets for multimodal therapies, the purpose of this study is to investigate the role of FGF8, ALK, and EML4 in upfront resected ESCC [18,19].

## 2. Materials and Methods

Patients who underwent upfront resection of ESCC between 1991 and 2011 at the Medical University of Vienna were identified from a prospectively maintained database. Patients with distant metastasis at the time of diagnosis or history of any other malignant disease were excluded. This study was approved by the Ethics Committee of the Medical University of Vienna, Borschkegasse 8b/E06, 1090 Vienna, reference number 1350/2015, according to the Declaration of Helsinki.

Immunohistochemistry (IHC) for FGF8, ALK, and EML4 expression was performed as described previously [7,20,21]. In brief, paraffin-embedded specimens fixed in 4% buffered formalin were used with 3 µm thick histological sections. Expression of FGF8 was detected using the polyclonal rabbit FGF8 antibody from Abcam^®^, Cambridge, UK, ab203030, in a dilution of 1:600. To determine ALK and EML4 expression, CONFIRM ALK01 (Ventana^®^, Cupertino, CA, USA, ready to use); NCL-ALK (Novocastra, Lecia Microsystems^®^, Wetzlar, Germany, 1:30); ALK D5F3 (Cell Signaling Technology^®^, Danvers, MA, USA, 1:250); EML4 monoclonal antibody (M01), clone 3C10, detecting amino acids 1–63 of human EML4 (Abcam^®^, Cambridge, UK, 1:100); and antibody NBP1-86805, covering, in EML4-001, amino acids 857–942, in EML4-002, amino acids 799–844, in EML4-003, amino acids 868–953, and in EML4-201, amino acids 121–206 (Novus Biologicals^®^, Littleton, CO, USA, 1:2000) were used, respectively. Immunostaining scores (0–12) of FGF8, ALK, and EML4 were calculated as the products of the staining intensity (0 = negative, 1 = weak, 2 = moderate, or 3 = strong expression) and points (0–4) were given for the percentages of tumor cells showing positive staining: 0 (<1%), 1 (1–10%), 2 (10–50%), 3 (51–80%), and 4 (>80%). Tumors were considered to have high expression if the final scores exceeded the median score. Tumors showing expression equal or below the median were considered as being low or absent.

Fluorescence in situ hybridization (FISH) was carried out as described previously [21]. Briefly, to investigate the ALK and EML4 gene status by FISH, a triple-color break-apart single-fusion probe (ZytoLight^®^ SPEC ALK/EML4 TriCheck™, ZytoVision, Bremerhaven, Germany) was used and analyzed according to the manufacturer’s instructions.

### Statistical Analysis

Statistical analysis was performed using the *R* statistical software, Vienna, Austria (version 3.6), with the survival package. The overall survival (OS) was defined as the time between primary surgery and the patient’s death. Death from causes other than ESCC or survival until the end of observation were considered as censored observations. Uni- and multivariable analyses were conducted using the Cox proportional hazard model as appropriate. *R* statistical software was used to perform the *log*-rank test to determine the significance of differences in survival times. The potential significance of correlations between clinicopathological factors and FGF8, ALK, and EML4 were analyzed with the *χ*^2^ test. Non-parametric Kendall’s rank correlation was performed to investigate potential statistical dependence between FGF8, ALK, and EML4 [22,23].

## 3. Results

In total, 122 patients with upfront resected ESCC were included in this study (Figure 1). IHC for FGF8 was successful in all patients whereas IHC and FISH for ALK and EML4 was successfully performed in 89 patients. The majority of patients (87, 71.3%) were male and had a moderately differentiated (G2) carcinoma (76, 62.3%). High expression of FGF8 was detected in 66 (54.1%) patients as compared to low/negative-expressing areas (Figure 2A,B, and Appendix A Table A2). While amplification of ALK and EML4 was found in nine (10.1%) and eight (9.0%) patients, respectively, no translocation could be observed in the eight-nine patients investigated (Figure 2C,D; C and D: usage with permission from Elsevier and Copyright Clearance Center (Schoppmann SF et al., 2013 [21])).

We found that all cases were negative for ALK and all cases were positive for EML4. Due to these uniform findings, no further data analysis regarding OS or correlations with other clinicopathological parameters was conducted. The results of the ALK and EML4 immunostainings demonstrated no correlation with the gene status as determined by FISH. Specifically, ALK immunostaining did not align with the presence or absence of ALK gene alterations, and similarly, EML4 protein expression was consistently strong, irrespective of the corresponding gene status. This suggests that the expression of these proteins, as observed in IHC experiments, was independent of their underlying genetic alterations.

A significant correlation was found between an overexpression of FGF8 and ALK (*p* = 0.04) and EML4 (*p* = 0.01) amplification, respectively. Clinicopathological characteristics and the correlations of FGF8, ALK, and EML4 for all patients can be found in Appendix A Table A1.

The mean time of follow-up was 54 ± 4 (standard error) months. The median OS was 18.44 months (range 0.3–290). Kaplan–Meier analysis indicated a significant correlation between high FGF8 expression and reduced patient OS (*p* = 0.02). In contrast, no significant associations were observed for ALK and EML4 amplification in relation to patient survival. Univariable Cox proportional hazard regression revealed that high FGF8 expression, gender, tumor grading, tumor stage, lymph node stage, and adjuvant therapy significantly impacted patient OS (Table 1).

In the multivariate Cox proportional hazard regression analysis, several clinicopathologic factors were identified as independent predictors of overall survival (OS) in patients with ESCC (Table 2). Male gender was associated with a significantly higher risk of mortality compared to females (*p* = 0.001) and age over 65 years was linked to worse OS compared to younger patients (*p* = 0.013). Tumor grading showed nearly significant differences, with grade 2 tumors carrying a worse prognosis than grade 3 (*p* = 0.046). Tumor stage was also a strong predictor, particularly for stage 2, which was associated with a higher risk of mortality compared to stage 4 (*p* < 0.001). No significant associations were found between lymph node involvement or resection status and OS. However, high FGF8 expression was an independent predictor of poor survival (*p* = 0.035).

## 4. Discussion

Globally, ESCC ranks among the primary causes of cancer-associated mortality [24]. Despite significant improvements in patient survival over the past several years, primarily due to the introduction of multimodal therapy strategies, OS rates remain low. Therefore, to create novel diagnostic tools and more effective treatment modalities, a deeper comprehension of the pathophysiology of ESCC is desperately needed [25]. Seven subfamilies make up the family of FGFs, which is divided based on functional characteristics, sequence similarity, and the secretion method. FGFs play a significant role in angiogenesis and cell proliferation. Heparin or heparin sulfate is required as a cofactor for the binding and activation of FGFRs by FGF8, FGF17, and FGF18, which make up the conventional FGF8 subfamily [26,27]. The FGF8 gene, located on chromosome 10q24.32, is involved in embryonic development and mediates the transitions from epithelial to mesenchymal and mesenchymal to epithelial. Additionally, FGF8 has a role in the development of the cardiovascular and cranopharyngeal regions. FGF and/or FGFR overexpression, FGFR gene amplification and fusion, or mutation may occur following changes in FGF signaling. Owing to these properties, FGF overexpression plays a crucial role in cancer by encouraging carcinogenesis and distant metastasis [5].

The closely spaced genes ALK and EML4 are found on chromosome 2′s short arm (2p21 and 2p23, respectively), separated by 12.7 megabases and oriented in opposing directions [28]. For several malignancies, one of the most significant molecular changes is the EML4-ALK fusion gene. Reverse transcription polymerase chain reaction (RT-PCR), in situ hybridization (FISH), and Ventana IHC (D5F3) are the recommended techniques for detecting ALK rearrangement. According to earlier research, the percentage of ALK-positive squamous cell carcinoma cases discovered by PCR or IHC/FISH is as low as 1% [29].

To date, studies investigating FGF8 in ESCC have relied primarily on cell line models, with limited clinical data derived from patient samples [30,31]. This underscores the relevance of our study, as it offers new perspectives by translating previous findings into a clinical context specific to ESCC. Furthermore, while our study suggests that FGF8 may serve as a prognostic marker, the evidence supporting its direct clinical utility is currently limited. Therapies targeting FGF8, ALK, or EML4 are not yet part of routine clinical practice. However, data from larger patient cohorts are urgently needed to validate these findings and assess their potential clinical impact.

The primary aim of our study was not solely to revisit the prognostic roles of these molecules but rather to explore a novel therapeutic angle and, specifically, overcome resistance to current therapies for ESCC. By examining the interplay between FGF8, ALK, and EML4, we aim to uncover new mechanisms that could enhance therapeutic efficacy. While the FGF8 pathway is indeed well documented in other cancers, its role in therapeutic resistance, particularly in ESCC, remains largely unexplored [32,33,34]. Despite progress, research on the therapeutic role of FGFs and FGFRs in ESCC remains in its early stages. Although several preclinical studies have identified targetable pathways, further substantial research is needed to fully clarify their clinical significance [21].

FGFR alterations are present in up to 40% of gastroesophageal cancers, often accompanied by other mutations that may confer resistance to FGF/R-targeted therapies [18,35]. The majority of targeted therapies have a recognized difficulty in overcoming acquired resistance. The most relevant mechanism of FGFR-TKI-acquired resistance is FGFR kinase mutation. Gatekeeper mutations have been found to cause resistance to FGFR inhibition in vitro by obstructing TKI access to the hydrophobic ATP binding site. The development of irreversible covalent FGFR inhibitors, like futibatinib, should persist, especially given the rise of gatekeeper mutations in FGFRs which contribute to resistance. According to a Japanese study, FGFR3 overexpression mediates ALK-inhibitor resistance in lung cancer [36]. This suggests that FGF/R-ALK signaling could be a future therapeutic target in ESCC, particularly to address acquired resistance.

In conclusion, while the research in this field has made progress, our study focuses on addressing these challenges by investigating the potential of FGF8 and ALK/EML4 pathways to provide a novel therapeutic approach to overcoming resistance in ESCC. By extending these pathways into clinical research for ESCC, our work provides insights that could be pivotal in improving therapeutic strategies for this aggressive cancer type.

## 5. Limitations

We must point out that our study is a single-center design, which may restrict the broader applicability of our findings. Furthermore, the modest size of our cohort, comprising 122 patients, limits the statistical power and generalizability of the results. However, given the rarity of clinical research in ESCC and the specific focus on FGF8, ALK, and EML4, this cohort provides a valuable foundation for initial insights. Future studies involving larger patient cohorts are warranted to enhance the robustness of these findings, potentially incorporating advanced techniques such as multiomics approaches and spatial profiling to enlighten the clinical significance of these biomarkers.

While fluorescence in situ hybridization (FISH) and immunohistochemistry (IHC) are well-established methodologies in both clinical and research settings, further validation is necessary to confirm the reproducibility and robustness of these findings in the context of ESCC. Despite the recognized reliability of these techniques in biomarker detection, additional studies with larger sample sizes are essential to strengthen the evidence supporting their application in this specific cancer type.

Moreover, we acknowledge the potential for bias introduced by the exclusion criteria and the patient selection process. Lastly, it is important to note that while our findings indicate a correlation between FGF8 expression and clinicopathological parameters, the lack of clinical data supporting the immediate use of FGF8, ALK, and EML4 as therapeutic targets underscores the need for further research in this area.

## 6. Conclusions

This is the first study that investigated the prognostic role of FGF8, ALK, and EML4 in upfront resected ESCC. FGF8 was found to be an independent prognostic factor for patients’ OS. Furthermore, a statistically significant correlation between the expression level of FGF8 and ALK and EML4 alteration could be demonstrated. Our results, in combination with previously published data describing FGF/R and ALK/EML4 interactions, might be the starting point for further investigations on the exact pathomechanisms of acquired TKI resistance in ESCC treatment.

## Figures and Tables

**Figure 1 cancers-16-03624-f001:**
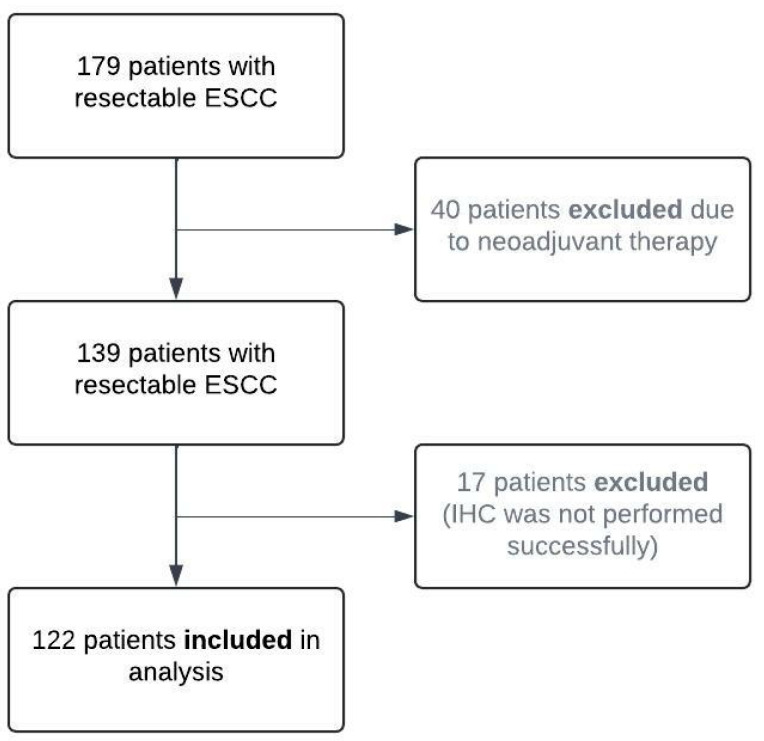
Study profile. ESCC indicates esophageal squamous cell carcinoma.

**Figure 2 cancers-16-03624-f002:**
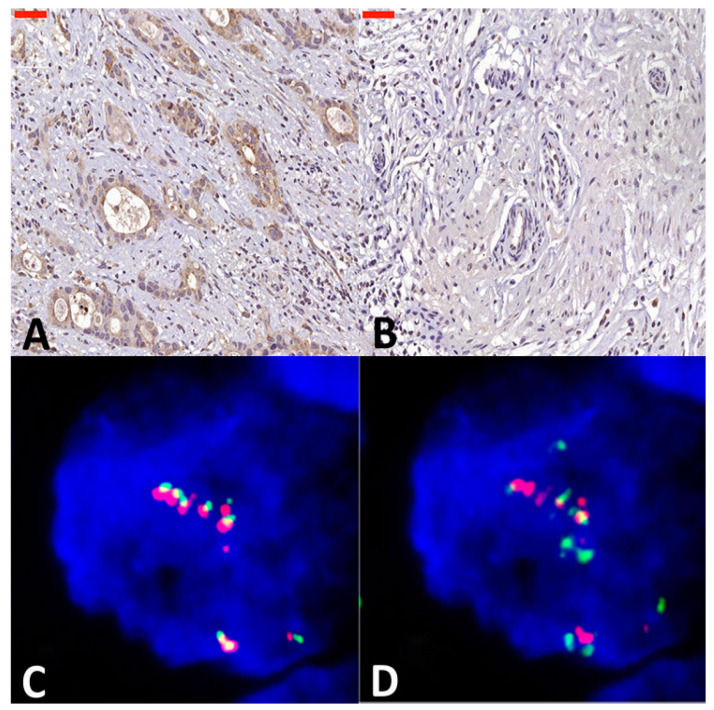
Representative specimen of esophageal squamous cell carcinoma showing (**A**) a high- and (**B**) a low-FGF8-expressing tumor section. The bar corresponds to 50 µm. Original magnification ×400 for both. (**C**) Esophageal squamous cell carcinoma with ALK gene amplification (red and green fusion signals flanking the ALK gene). (**D**) Specimen identical as C showing EML4 amplification in turquoise. (C and D: usage with permission from Elsevier and Copyright Clearance Center (Schoppmann SF et al., 2013 [21])).

**Table 1 cancers-16-03624-t001:** Univariate Cox regression analyses estimating the influence of FGF8 expression—ALK and EML4 alteration and clinicopathologic parameters on OS of patients with ESCC.

Factors		HR	CI (95%)	*p*-Value
FGF8 (ref. low/negative)		1.58	1.077–2.319	0.018
ALK amplification (ref. no)		1.839	0.875–3.865	0.136
EML4 amplification (ref. no)		1.791	0.816–3.930	0.177
Age		1.003	0.984–1.023	0.738
Sex (ref. male)		2.41	1.517–3.827	<0.001
Grading				
	1 vs. 3	2.561	1.272–5.159	0.008
	2 vs. 3	2.638	1.253–5.555	0.011
UICC				
	1 vs. 4	2.273	1.171–4.412	0.015
	2 vs. 4	3.127	1.654–5.913	<0.001
	3 vs. 4	3.74	1.715–8.155	<0.001
pT				
	1 vs. 4	2.16	1.144–4.076	0.017
	2 vs. 4	3.751	2.108–6.675	<0.001
	3 vs. 4	1.75	0.676–4.531	0.249
pN				
	1 vs. 0	1.75	1.131–2.708	0.012
	2 vs. 0	2.206	1.290–3.774	0.004
	3 vs. 0	2.036	0.907–4.572	0.085
R (ref. R0)		1.506	0.896–2.531	0.14
Adjuvant therapy (ref. no)		2.061	1.404–3.024	<0.001

FGF8 = fibroblast growth factor 8; ALK = anaplastic lymphoma kinase; EML4 = echinoderm microtubule-associated protein-like 4; UICC = Union Internationale contre le Cancer; pT = pathological tumor stage; pN = pathological lymph node stage; R = resection margin.

**Table 2 cancers-16-03624-t002:** Multivariate Cox regression analyses estimating the influence of FGF8 expression and clinicopathologic parameters on OS of patients with ESCC.

Factors		HR	CI (95%)	*p*-Value
Sex (ref. male)		2.276	1.393–3.720	0.001
Age65 (ref. <65)		1.686	1.118–2.543	0.013
G				0.136
	1 vs. 3	1.906	0.919–3.953	0.083
	2 vs. 3	2.221	1.014–4.864	0.046
pT				0.001
	1 vs. 4	1.874	0.961–3.654	0.065
	2 vs. 4	3.343	1.774–6.302	0.000
	3 vs. 4	2.154	0.720–6.448	0.170
pN				0.316
	1 vs. 0	1.334	0.821–2.167	0.245
	2 vs. 0	0.780	0.412–1.475	0.444
	3 vs. 0	1.290	0.546–3.048	0.562
R (ref. 0)		1.107	0.580–2.115	0.758
FGF8 (high)		1.605	1.033–2.494	0.035

FGF8 = fibroblast growth factor 8; pT = pathological tumor stage; pN = pathological lymph node stage.

## Data Availability

Data are not available due to ethical restrictions.

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
