# Peer review of "Overexpression of Fibroblast Growth Factor 8 Is a Predictor of Impaired Survival in Esophageal Squamous Cell Carcinoma and Correlates with ALK/EML4 Alteration"

_cancers, 2024, doi:10.3390/cancers16213624_

Round 1

Reviewer 1 Report

Comments and Suggestions for Authors

The manuscript titled "Overexpression of Fibroblast Growth Factor 8 is a Predictor of Impaired Survival in Esophageal Squamous Cell Carcinoma and Correlates with ALK/EML4 Alteration" investigates the prognostic role of Fibroblast Growth Factor 8 (FGF8) together with ALK and EML4 gene alterations in esophageal squamous cell carcinoma (ESCC) by analyzing tissue samples from 122 patients who underwent surgical resection. The authors suggest that FGF8 may serve as a potential biomarker to predict patient outcomes and may be a therapeutic target in ESCC, but further research is needed to confirm these findings. However, there are some issues and questions that should be addressed in the study.

1- The study's focus on the prognostic role of FGF8, ALK, and EML4 in ESCC lacks novelty. The manuscript does not provide sufficient new insights or significantly advance the current understanding of ESCC beyond what is already known in the literature. The potential role of FGF8 and related pathways in cancer prognosis is a well-documented area, and the manuscript does not present any particularly new approaches or findings.

2- The methods used for detection and analysis of biomarkers (e.g. FISH and IHC) are standard techniques; however, the article lacks details regarding the robustness of the methods and reproducibility of the findings. Additionally, the study cohort was relatively small (122 patients), limiting the statistical power and generalizability of the results. The exclusion criteria and patient selection process may also introduce bias that was not adequately addressed.

3- Although the manuscript suggests that FGF8 may serve as a prognostic marker, there is limited evidence to support its clinical utility. The study does not provide convincing data to show how FGF8 expression would alter clinical management or patient outcomes. The potential therapeutic implications discussed are speculative and are not supported by robust preclinical or clinical evidence.

4- The manuscript does not adequately control for confounding variables that may affect survival outcomes, such as differences in treatment regimens, comorbidities, or differences in tumor biology in the patient cohort. The multivariate analysis presented does not adequately address these potential confounders, raising concerns about the validity of the results.

5- The Discussion section does not comprehensively review the implications of the findings, limitations of the study, or possible directions for future research. Furthermore, the lack of critical engagement with the existing literature weakens the scholarly impact of the manuscript.

Comments on the Quality of English Language

Moderate editing of English language required.

Author Response

Reviewer #1

  1. “The study's focus on the prognostic role of FGF8, ALK, and EML4 in ESCC lacks novelty. The manuscript does not provide sufficient new insights or significantly advance the current understanding of ESCC beyond what is already known in the literature. The potential role of FGF8 and related pathways in cancer prognosis is a well-documented area, and the manuscript does not present any particularly new approaches or findings.”

We thank the reviewer for this valuable feedback. We appreciate and recognize the concerns regarding the novelty of the prognostic focus on FGF8, ALK, and EML4 in ESCC. However, we would like to emphasize that, in contrast to other malignancies, data on FGF8 in ESCC is still very scarce. To date, studies investigating FGF8 in this cancer type have relied on cell line models, with very little clinical data derived from patient samples (Awan AK, et al., “Androgen receptors may act in a paracrine manner to regulate oesophageal adenocarcinoma growth”, EJSO 33 (2007) 561e568; and Tanaka S, et al., “A Novel Isoform of Human Fibroblast Growth Factor 8 Is Induced by Androgens and Associated with Progression of Esophageal Carcinoma”, Digestive Diseases and Sciences, Vol. 46, No. 5 (May 2001), pp. 1016–1021). This makes our study particularly relevant, as it contributes new insights by extending the research into a clinical context for ESCC.

Furthermore, the focus of our study was not solely to revisit the prognostic roles of these molecules but rather to explore a novel therapeutic angle—specifically, overcoming resistance to current state-of-the-art therapies for ESCC. By examining the interplay between FGF8, ALK, and EML4, we aim to uncover new mechanisms that could enhance therapeutic efficacy. While the FGF8 pathway is indeed well-documented in other cancers, its role in therapeutic resistance, particularly in ESCC, remains largely unexplored.

Additionally, we would like to stress that this manuscript is intended for the Special Issue “Oncology: State-of-the-Art Research in Austria”, which highlights current developments in Austrian cancer research. Our study aligns with this aim by investigating new potential strategies in the Austrian context for improving ESCC outcomes, a cancer with distinct challenges and unmet clinical needs in this region. Therefore, we believe the work significantly contributes to the state-of-the-art research in ESCC, offering innovative insights for both Austrian and global oncology research.

The changes regarding this comment, are incorporated in the Discussion section and highlighted in the revised manuscripts version.

  1. “The methods used for detection and analysis of biomarkers (e.g. FISH and IHC) are standard techniques; however, the article lacks details regarding the robustness of the methods and reproducibility of the findings. Additionally, the study cohort was relatively small (122 patients), limiting the statistical power and generalizability of the results. The exclusion criteria and patient selection process may also introduce bias that was not adequately addressed.”

We thank the reviewer for this insightful feedback. We appreciate the comments regarding the detection and analysis methods, such as FISH and IHC, used in our study. These techniques are indeed well-established and routinely employed in both clinical and research settings, which ensures a high degree of reliability and validity in the results. The standard use of these methodologies supports their robustness and accuracy for biomarker detection, including FGF8, ALK, and EML4, in this context.

Furthermore, as noted in Reviewer#1s first comment, data regarding the prognostic role of FGF8, ALK, and EML4 in ESCC, while limited, already exists, and our study builds upon these foundations. Specifically, the reproducibility of our findings can be supported by further studies published by our study group (Schoppmann SF, et al., “Amplification but not translocation of anaplastic lymphoma kinase is a frequent event in oesophageal cancer.”, Eur J Cancer. 2013 May;49(8):1876-81; Preusser M, et al., “ALK gene translocations and amplifications in brain metastases of non-small cell lung cancer.”, Lung Cancer. 2013 Jun;80(3):278-83.; Berghoff AS, et al., “ALK gene aberrations and the JUN/JUNB/PDGFR axis in metastatic NSCLC.”, APMIS. 2014 Sep;122(9):867-72.; Preusser M, et al., “Spectrum of gene mutations detected by next generation exome sequencing in brain metastases of lung adenocarcinoma.”, Eur J Cancer. 2015 Sep;51(13):1803-11.).

This work provides a solid basis for the use of these biomarkers in ESCC research. Their research reinforces the methodological consistency and strengthens the validity of our approach in evaluating these biomarkers.

In light of these points, we believe our methodology ensures both reliability and reproducibility, while addressing the current gaps in ESCC-specific data on these markers.

Furthermore, we acknowledge the reviewers comment that the relatively small cohort of 122 patients in our study may limit the statistical power and generalizability of the results. However, given the rarity of ESCC cases and the specific focus on FGF8, ALK, and EML4, we believe this cohort provides a solid foundation for initial insights.

We fully agree that larger patient collectives would enhance the robustness of the findings, and we are planning further studies to address this limitation. These future investigations will involve a larger patient cohort and incorporate advanced techniques such as multiomics approaches and spatial profiling. This will allow for a more comprehensive understanding of the molecular landscape of ESCC and help validate the clinical relevance of FGF8, ALK, and EML4 in therapeutic resistance and prognosis. We believe these efforts will significantly strengthen the generalizability and clinical applicability of our findings.

Furthermore, we acknowledge the concern regarding potential bias introduced by the exclusion criteria and patient selection process. In response to this, we have added a detailed study profile in the revised version of the manuscript, which outlines the selection process and exclusion criteria more clearly. We believe this addition addresses the concern by providing greater transparency and a more thorough explanation of how patients were selected for the study. We hope that with this added information, any concerns about potential bias will be mitigated.

The changes regarding this comment, are incorporated and highlighted in the revised manuscripts version.

  1. “Although the manuscript suggests that FGF8 may serve as a prognostic marker, there is limited evidence to support its clinical utility. The study does not provide convincing data to show how FGF8 expression would alter clinical management or patient outcomes. The potential therapeutic implications discussed are speculative and are not supported by robust preclinical or clinical evidence.”

Thank the reviewer for this thoughtful feedback. We fully acknowledge that while our study suggests FGF8 may serve as a prognostic marker, the evidence supporting its direct clinical utility is currently limited. It is correct that therapies targeting FGF8, ALK, or EML4 are not yet part of routine clinical practice. However, as mentioned by Reviewer#1s first remark, data from larger patient cohorts are urgently needed to validate these biomarkers and assess their potential clinical impact.

Moreover, the primary aim of our study was not to establish immediate clinical applicability but to investigate the potential of these biomarkers in overcoming resistance to state-of-the-art therapies for ESCC. Our findings represent an important early step in identifying novel therapeutic targets, and we agree that further preclinical and clinical studies will be essential to substantiate the therapeutic implications. We believe this exploratory work lays the groundwork for future research with larger cohorts and more robust data to assess the full clinical potential of these biomarkers.

The changes regarding this comment, are incorporated and highlighted in the revised manuscripts version.

  1. “The manuscript does not adequately control for confounding variables that may affect survival outcomes, such as differences in treatment regimens, comorbidities, or differences in tumor biology in the patient cohort. The multivariate analysis presented does not adequately address these potential confounders, raising concerns about the validity of the results.”

Based on this high-quality remark, we have thoroughly reevaluated and overhauled the multivariate Cox model to address potential confounding variables more effectively. Specifically, we have removed the UICC staging factor due to its redundancy with pT and pN, and we excluded adjuvant therapy from the model, as this factor is not known at the time of surgery. To improve the robustness of the analysis, we have added gender and age as variables.

We believe that this revised multivariate model now provides a more accurate and reliable analysis, addressing the concerns about potential confounders and enhancing the validity of our results. Thank you for bringing this to our attention, and we hope the new approach better reflects the true impact of the biomarkers studied.

These changes are highlighted in the revised manuscripts version.

  1. “The Discussion section does not comprehensively review the implications of the findings, limitations of the study, or possible directions for future research. Furthermore, the lack of critical engagement with the existing literature weakens the scholarly impact of the manuscript.”

We are thankfull for this constructive feedback. We agree that the Discussion section could benefit from a more thorough exploration of the study's implications, limitations, and future research directions. In response, we have expanded the Discussion in the revised manuscript to include a more comprehensive review of our findings and their potential clinical relevance. We have also added a detailed discussion of the study's limitations, such as the relatively small sample size, the need for larger patient cohorts, and the absence of clinical data supporting the immediate use of FGF8, ALK, and EML4 as therapeutic targets.

Moreover, we have critically engaged with additional relevant literature to better contextualize our findings within the broader field of ESCC research. We hope that this enhanced discussion not only strengthens the scholarly impact of the manuscript but also provides clearer directions for future research, particularly in the areas of therapeutic resistance and biomarker validation through multiomics and larger-scale studies. Thank you for highlighting these important areas for improvement.

These changes are implemented and highlighted in the revised manuscripts version.

Reviewer 2 Report

Comments and Suggestions for Authors

The manuscript: "Overexpression of Fibroblast Growth Factor 8 is a Predictor of Impaired Survival in Esophageal Squamous Cell Carcinoma and Correlates with ALK/EML4 Alteration" by Gerd Jomrich et al., presents an investigation of FGF8, ALK, and EML4 proteins role on the prognosis of esophageal squamous cell carcinoma (ESCC) patients. The conclusions are based on IHC and FISH experiments results on resected tumor tissues from 122 patients and the analyses of their clinicopathological data. The corelation of data was conducted using uni- and multivariable Cox proportional hazard model analyses. The authors concluded that FGF8 overexpression is an independent prognostic factor for patients’ survival and that is a statistically significant correlation between the expression level of FGF8 and ALK and EML4 genes amplification.
Although the finding of biomarkers for prognosis and targeted therapy in Esophageal Squamous Cell Carcinoma is necessary, the present manuscript is lacking scientific soundness regarding the presentation of results, the disscusion  and consequently the conclusion.

Major comments:
1.    The authors claimed that samples tissues resected from 122 patients were analysed by IHC or by FISH. All samples were successful performed for IHC staining of FGH8 and 89 samples were successful performed for IHC and FISH of ALK and EML4.
However, no images showing the ALK and EML4 protein staining of tissues sample was provided. Moreover, no discussion about the results obtained for IHC experiments of ALK and EML4 could be found in Results and Discussion sections even that the authors mentioned that 6 different antibodies were used (for ALK and EML4) in Material and Methods Section (line 104-111)??? Please provide the images of ALK and EML4 IHC results, and add the discussion of the results in Results section.
Moreover, the authors wrote that the FISH images can not be provided due to copyright reasons?? I not understand this? Please explain!
2.    Also, the authors wrote that all IHC samples were analyzed and scored based on Intensity of Staining and Percentage of Positive Cells, and the results were translated in a system of scoring ranging from 0 to 12. The results for all stainings reflected in this scoring system should be provide as a tabel or graph for all three analyzed proteins.
3.    The authors should discuss the results of ALK and EML4 immunostainings and FISH. There is no indication in manuscript about the existence or lack of corelation between proteins expression (IHC experiments) and their gene status (FISH)!!
4.    Line 202-203: “to earlier research, the percentage of ALK-positive squamous cell carcinoma cases discovered by PCR or IHC/FISH is as low as 1%.” The reference is missing
5.    In my opinion, the manuscript should be rewritten and the authors should complete the results, discussion and conclusion sections with the necessary information to be scientifically argued. The Results section should also include images from the IHC and FISH experiments for ALK and EML4, as well as a table with the IHC scores for all 3 proteins.

Author Response

  1. “The authors claimed that samples tissues resected from 122 patients were analysed by IHC or by FISH. All samples were successful performed for IHC staining of FGH8 and 89 samples were successful performed for IHC and FISH of ALK and EML4. However, no images showing the ALK and EML4 protein staining of tissues sample was provided. Moreover, no discussion about the results obtained for IHC experiments of ALK and EML4 could be found in Results and Discussion sections even that the authors mentioned that 6 different antibodies were used (for ALK and EML4) in Material and Methods Section (line 104-111)??? Please provide the images of ALK and EML4 IHC results, and add the discussion of the results in Results section. Moreover, the authors wrote that the FISH images can not be provided due to copyright reasons?? I not understand this? Please explain!”

We totally agree with the reviewers remark that the results regarding the IHC analysis of ALK and EML4 were not detailed enough in the original manuscript. Upon review, we found that all cases were negative for ALK and all cases were positive for EML4. Due to these uniform findings, no further data analysis regarding overall survival or correlations was conducted, which we now realize was not clearly communicated in the original submission.

In response to your comments, we have revised the manuscript to include a clearer presentation of these IHC results, along with a discussion explaining why no additional analysis was performed. We hope that this addresses your concerns and provides a more thorough and transparent presentation of the data. Thank you for pointing this out, and we believe the revisions now make these results clearer and more complete.

Furthermore, we thank the reviewer for the question regarding the FISH images. We apologize for any confusion caused. In the figure legends, we clearly stated that we have obtained the necessary permission to use the images, and we can certainly provide them in the revised manuscript.

These changes are highlighted in the revised manuscripts version.

  1. “Also, the authors wrote that all IHC samples were analyzed and scored based on Intensity of Staining and Percentage of Positive Cells, and the results were translated in a system of scoring ranging from 0 to 12. The results for all stainings reflected in this scoring system should be provide as a tabel or graph for all three analyzed proteins.”

As proposed by the reviewer, detailed information regarding the scoring system used for immunohistochemical results of FGF8 are added to new manuscripts version (supplement). Due to the results mentioned in the response to reviewer #2's first comment (IHC analysis: all cases were negative for ALK and all cases were positive for EML4), these data are not presented in additional form.

These changes are highlighted in the revised manuscripts version and can be found in the supplementary section.

  1. “The authors should discuss the results of ALK and EML4 immunostainings and FISH. There is no indication in manuscript about the existence or lack of corelation between proteins expression (IHC experiments) and their gene status (FISH)!!”

Due to the results mentioned in the response to reviewer #2's first comment (IHC analysis: all cases were negative for ALK and all cases were positive for EML4), the results of the ALK and EML4 immunostainings in our study demonstrated no correlation with the gene status as determined by FISH. Specifically, ALK immunostaining did not align with the presence or absence of ALK gene alterations, and similarly, EML4 protein expression was consistently strong, irrespective of the corresponding gene status. This suggests that the expression of these proteins, as observed in IHC experiments, was independent of their underlying genetic alterations.

These changes made in the Results and Discussion section are highlighted in the revised manuscripts version.

  1. “Line 202-203: “to earlier research, the percentage of ALK-positive squamous cell carcinoma cases discovered by PCR or IHC/FISH is as low as 1%.” The reference is missing”

Thanks to the prudent reviewer, the missing reference was added. (Wang H, et al. A study of ALKpositive pulmonary squamouscell carcinoma: from diagnostic methodologies to clinical efficacy. Lung Cancer. 2019;130:135–42.)

These changes are highlighted in the revised manuscripts version.

  1. “In my opinion, the manuscript should be rewritten and the authors should complete the results, discussion and conclusion sections with the necessary information to be scientifically argued. The Results section should also include images from the IHC and FISH experiments for ALK and EML4, as well as a table with the IHC scores for all 3 proteins.”

We really appreciate the reviewers profound feedback and for highlighting the key points that are most critical to the evaluation of our manuscript. We understand that there may be additional concerns, and we are committed to addressing each issue thoroughly to improve the quality of our work. Your insights regarding the most important aspects have been especially helpful, and we will prioritize those in our revisions. We hope that the changes based on the high-quality “Remarks 1-5” have now resulted in a significant improvement of the manuscript and thus made an evaluation possible.

Reviewer 3 Report

Comments and Suggestions for Authors
  •  
    • This study explores the association between FGF8 and EML4-ALK fusion genes in esophageal squamous cell carcinoma, which has certain clinical application value and novelty. Upregulation of FGF8 expression can be observed in several malignant tumors such as colorectal cancer and alveolar rhabdomyosarcoma. This study requires additional evidence before it can be published. The immunofluorescence image in Figure 1 is unclear and references other literature. Could you please supplement this part of the results. Further verification of the correlation between FGF8 and EML4-ALK fusion genes at the cellular and animal levels is needed to further validate the conclusions of this study.
  •  
  •  
  •  
  •  
  •  
  •  
  •  
  •  
  • 10.1016/j.ejso.2006.12.001

Author Response

  1. “This study explores the association between FGF8 and EML4-ALK fusion genes in esophageal squamous cell carcinoma, which has certain clinical application value and novelty. Upregulation of FGF8 expression can be observed in several malignant tumors such as colorectal cancer and alveolar rhabdomyosarcoma. This study requires additional evidence before it can be published. The immunofluorescence image in Figure 1 is unclear and references other literature. Could you please supplement this part of the results. Further verification of the correlation between FGF8 and EML4-ALK fusion genes at the cellular and animal levels is needed to further validate the conclusions of this study.

10.1016/j.ejso.2006.12.001”

The reviewers remarks addresses highly important issues and we thank the reviewer for this thoughtful feedback on the novelty and clinical relevance of our study exploring the association between FGF8 and EML4-ALK fusion genes in esophageal squamous cell carcinoma. We greatly appreciate this recognition of its potential value.

Regarding the image in Figure 1, we apologize for any confusion caused. In the figure legends, we clearly stated that we obtained the necessary permission to use the images. We can certainly include these permissions more clearly in the revised manuscript and ensure the images are presented in the highest quality.

As for the need for further validation of the correlation between FGF8 and EML4-ALK fusion genes, we completely agree. While our current findings indicate that ALK and EML4 immunostainings were independent of gene status, more advanced investigations are indeed essential. We recognize that extending these studies to animal models and incorporating methods such as spatial profiling and multiomics would provide more comprehensive insights. Our current results, however, provide a solid foundation to build upon with these further experiments.

These changes are highlighted in the revised manuscripts version.

Round 2

Reviewer 1 Report

Comments and Suggestions for Authors

I am satisfied that the authors have addressed all of my previous concerns about the article. It is now much improved and I feel that it is now suitable for publication.

Comments on the Quality of English Language

OK

Reviewer 2 Report

Comments and Suggestions for Authors

The authors responded to all comments. The manuscript could be published in this form.

Reviewer 3 Report

Comments and Suggestions for Authors

The author did not provide additional data to support the conclusion